# Association of Alternative Markers of Carbohydrate Metabolism (Fructosamine and 1,5-Anhydroglucitol) with Perioperative Characteristics and In-Hospital Complications of Coronary Artery Bypass Grafting in Patients with Type 2 Diabetes Mellitus, Prediabetes, and Normoglycemia

**DOI:** 10.3390/diagnostics13050969

**Published:** 2023-03-03

**Authors:** Alexey N. Sumin, Natalia A. Bezdenezhnykh, Andrey V. Bezdenezhnykh, Anastasiya A. Kuzmina, Yuliya A. Dyleva, Olga L. Barbarash

**Affiliations:** Research Institute for Complex Issues of Cardiovascular Diseases, Sosnovy Blvd. 6, Kemerovo 650002, Russia

**Keywords:** coronary bypass surgery, carbohydrate metabolism disorders, postoperative complications, fructosamine, 1,5-anhydroglucitol

## Abstract

Patients with type 2 diabetes make up 25 to 40% of those referred for coronary bypass surgery, and the impact of diabetes on the results of the operation is studied in various aspects. To assess the state of carbohydrate metabolism before any surgical interventions, including CABG, daily glycemic control, and the determination of glycated hemoglobin (HbA1c) is recommended. Glycated hemoglobin reflects the glucose concentration for the 3 months prior to the measurement, but alternative markers that reflect glycemic fluctuations over a shorter period of time may be useful in preoperative preparation. The aim of this study was to study the relationship between the concentration of alternative markers of carbohydrate metabolism (fructosamine and 1,5-anhydroglucitol) with patients’ clinical characteristics and the rate of hospital complications after coronary artery bypass grafting (CABG). Method. In the cohort of 383 patients, besides the routine examination, additional markers of carbohydrate metabolism were determined before and on days 7–8 after CABG: glycated hemoglobin (HbA1c), fructosamine, and 1,5-anhydroglucitol. We evaluated the dynamics of these parameters in groups of patients with diabetes mellitus, prediabetes, and normoglycemia, as well as the association of these parameters with clinical parameters. Additionally, we assessed the incidence of postoperative complications and factors associated with their development. Results. In all groups of patients (diabetes mellitus, prediabetes, normoglycemia), there was a significant decrease in the level of fructosamine on the 7th day after CABG compared with baseline (p1st–2nd point 0.030, 0.001, and 0.038 in groups 1, 2, and 3, respectively), whereas the level of 1,5-anhydroglucitol did not change significantly. The preoperative level of fructosamine was associated with the risk of surgery according to the EuroSCORE II scale (*p* = 0.002), as were the number of bypasses (*p* = 0.012), body mass index and overweightness (*p* < 0.001 in both cases), triglyceride (*p* < 0.001) and fibrinogen levels (*p* = 0.002), preoperative and postoperative glucose and HbA1c levels (*p* < 0.001 in all cases), left atrium size (*p* = 0.028), multiplicity of cardioplegia, cardiopulmonary bypass duration and aortic clamp time (*p* < 0.001 in all cases). The preoperative level of 1,5-anhydroglucitol showed an inverse correlation with fasting glucose and fructosamine levels before surgery (*p* < 0.001 in all cases), intima media thickness (*p* = 0.016), and a direct correlation with LV end-diastolic volume (*p* = 0.020). The combined endpoint (presence of significant perioperative complications + extended hospital stay after surgery >10 days) was identified in 291 patients. In binary logistic regression analysis patient age (*p* = 0.005) and fructosamine level (*p* = 0.022) were independently associated with the development of this composite endpoint (presence of significant perioperative complications + extended hospital stay after surgery >10 days). Conclusions: This study demonstrated that in patients after CABG there was the significant decrease in the level of fructosamine compared with baseline, whereas the level of 1,5-anhydroglucitol did not change. Preoperative fructosamine levels were one of the independent predictors of the combined endpoint. The prognostic value of preoperative assessment of alternative markers of carbohydrate metabolism in cardiac surgery deserves further study.

## 1. Introduction

Coronary artery bypass grafting (CABG) is the best method of myocardial revascularization for patients with diabetes mellitus (DM) and multivessel coronary disease [1]. Among patients undergoing CABG, the proportion of patients with DM is constantly growing and currently reaches 40%; the presence of DM increases the number of postoperative complications and worsens the long-term prognosis [2,3,4]. Therefore, the search for ways to reduce the negative impact of DM on the results of coronary bypass surgery continues, the optimal targets for carbohydrate metabolism are being studied, and methods of preoperative preparation and perioperative management of patients with DM are being improved [3,5].

Currently, to assess the state of carbohydrate metabolism before CABG, the determination of glycated hemoglobin (HbA1c) is recommended [5]. Glycated hemoglobin reflects the concentration of glucose throughout the life of an erythrocyte, i.e., 3 months prior to the measurement. During preoperative preparation, alternative markers may be useful, reflecting glycemic fluctuations over a shorter period [6]. Markers of carbohydrate metabolism such as fructosamine and 1,5-anhydroglucitol are deprived of these restrictions [7]. 

Fructosamines are called glycated blood serum proteins formed during the reaction of glucose mainly with albumin [8]. The half-life of serum proteins is less than the life of red blood cells. Therefore, unlike glycated hemoglobin, the level of fructosamine reflects the degree of permanent or transient increase in glucose levels not in 3 months but in 1–3 weeks prior to the study. Recently, publications have begun to appear on the relationship of this marker with cardiovascular prognosis [6,9]. Since fructosamine may provide a more accurate assessment of glycemic variability and short-term therapeutic efficacy than HbA1c [7], it may be useful in assessing the achievement of carbohydrate metabolism compensation in preparing patients with DM for coronary bypass surgery. However, until now, fructosamine has not been not practically used for this purpose and there are only a few studies on this topic [10].

1,5-Anhydroglucitol (1,5-AG) is a glucid molecule, and tubular reabsorption of 1,5-AG competes with glucose. In situations where the glucose concentration exceeds the renal threshold of approximately 180 mg/dL (10 mmol/L), glomerular glucose excretion increases, as does its tubular reabsorption. In this situation, 1,5-AG, normally filtered in the glomerulus, is not reabsorbed into the tubules, increasing its urinary excretion and decreasing plasma concentration. Therefore, the plasma concentration of 1,5-AG may be a marker of prior (1–2 weeks) exposure to hyperglycemia above the renal glucose threshold, reflecting postprandial hyperglycemia peaks [7]. 1,5-AG is a biomarker for acute hyperglycemia; in acute hyperglycemia, renal reabsorption is inhibited by glucose and 1,5-AG is excreted in the urine, whereas its serum level decreases rapidly. 1,5-AG reflects jumps in glucose levels from 1–3 days to 2 weeks [11]. In this regard, a low level of serum 1,5-AG may be a clinical marker of short-term glycemic disorders and low levels of 1,5-AG reflect severe plaque calcification in CAD [12] and may be a predictor of cardiovascular disease and mortality after acute coronary syndrome [13]. With planned PCI, low and exacerbated levels of 1,5-anhydroglucitol are associated with cardiovascular events [12,14]; however, this biomarker has not been studied in cardiac surgeries.

The aim of this study was to study the relationship between the concentration of alternative markers of carbohydrate metabolism (fructosamine and 1,5-anhydroglucitol) with patients’ clinical characteristics and the rate of hospital complications after CABG.

## 2. Subjects, Materials, and Methods

### 2.1. Study Population

This single-center, cross-sectional, observational study was conducted at the Research Institute for Complex Issues of Cardiovascular Diseases, Kemerovo. Consecutive patients who under-went elective CABG in the cardiovascular surgery department of the clinic from 22 March 2011 to 22 March 2012 were included. The study design is shown in Figure 1. In total, the study involved 732 consecutive patients who were planned for CABG. For 9 of them, the revascularisation tactics were revised to percutaneous intervention due to the comorbidity, and 15 patients were denied myocardial revascularization and were excluded from the study. Thus, CABG was performed in 708 patients included in the study. In 383 consecutive patients, fructosamine and 1,5-anhydroglucitol were determined. Upon admission to the hospital for preparation for CABG in all of these 383 patients, glycemic status was examined. 

For patients with borderline fasting hyperglycemia according to the criteria of the World Health Organization/International Diabetes Federation (6.1–6.9 mmol/L (110–125 mg/dL) and without previously established diabetes mellitus, as well as patients with previously known prediabetes and in the absence of contraindications, an oral glucose tolerance test with 75 g of glucose was performed. If the results of several fasting studies or postprandial glycemia was sufficient to establish a diagnosis of diabetes, an oral glucose tolerance test was not performed. The diagnosis of type 2 diabetes mellitus and other disorders of carbohydrate metabolism (CMD) was established in accordance with current WHO criteria for the modern classification of diabetes mellitus and other glycemic disorders [15,16]. 

### 2.2. Data Collection

Baseline preoperative and perioperative indicators were obtained from the electronic database of the CABG registry and medical records. The data of anamnesis, laboratory examinations, echocardiography, coronary angiography, ultrasound and angiographic examination of the aorta, brachiocephalic and peripheral arterial pools, and the frequency of postoperative complications were analyzed. Confirmation of the presence and assessment of the prevalence of atherosclerotic lesions was carried out using color duplex scanning of the extracranial sections of the brachiocephalic arteries and arteries of the lower extremities (Aloka 5500 device). Not earlier than six months before CABG, patients underwent coronary angiography. Stenosis of the main coronary arteries, narrowing the lumen of the vessel by 70% or more or the trunk of the left coronary artery by 50% or more, was considered hemodynamically significant. A more detailed description of the materials and methods of this study, including criteria for the diagnosis of diabetes mellitus and other glycemic disorders, as well as preoperative examination and patient preparation and perioperative glycemic management, are described in a previously published article [17].

### 2.3. Evaluation of Indicators of Carbohydrate Metabolism 

The concentration of glucose in venous blood plasma was assessed by the hexokinase method. The level of fructosamine was determined by the kinetic colorimetric method; the level of 1,5-anhydrogluccitol (1.5 AG) was determined by enzyme immunoassay. An increase in fructosamine corresponds to an increase in glucose levels, and an increase in 1,5-anhydroglucitol corresponds to a decrease in glycemia. Clear reference values for these markers have not been established. The level of glycated hemoglobin (HbA1c) of hemolyzed whole blood was determined by turbidimetric inhibitory immunoassay according to the NGSP (National Glycohemoglobin Standardization Program) standard and standardized according to the reference values adopted in the Diabetes Control and Complications Trial (DCCT). A level of HbA1c up to 6.0% (42 mmol/mol) was considered normal.

### 2.4. Hospital Postoperative Complications

We took into account the following perioperative complications of CABG: the development of intra- and postoperative myocardial infarction; heart failure requiring prolonged inotropic support; paroxysms of atrial fibrillation; stroke; acute renal failure; multiple organ failure; respiratory complications (pneumonia, respiratory failure, hydrothorax); complications from the wound of the sternum (prolonged exudation, purulent complications, diastasis of the sternum, mediastinitis); bleeding and remediastinotomy for bleeding. An analysis of hospital mortality and its causes was also carried out. Two combined endpoints were used to analyze hospital treatment outcomes: (1) presence of significant perioperative complications + prolonged hospital stay after surgery (>10 days); and (2) presence of significant perioperative complications. All complications described above were considered significant, with the exception of the following: hydrothorax, pneumothorax, and hydropericardium not requiring puncture and wound complications that do not require secondary surgical treatment of the wound, antibiotic therapy, and an increase in the length of stay in the hospital.

### 2.5. Statistical Analyses

Statistical processing was carried out using standard software packages “STATISTICA 8.0” (Dell Software, Inc., Round Rock, TX, USA) and SPSS 17.0 (IBM, Armonk, NY, USA). The distribution of quantitative data was checked using the Shapiro–Wilk test. Because the distribution of all quantitative characteristics differed from normal, they were described using the median, indicating the upper and lower quartiles (25th and 75th percentiles). To compare three independent groups, the Kruskal–Wallis test was used, followed by analysis of intergroup differences using the Mann–Whitney method or the χ2 test. The Wilcoxon test was used to assess the perioperative dynamics of carbohydrate metabolism. For a small number of observations, the Fisher’s exact test with Yates correction was used. To solve the problem of multiple comparisons, the Bonferroni correction was used. Thus, taking into account the number of degrees of freedom, the critical level of significance *p* when comparing the three groups was taken equal to 0.017; in other cases-it was 0.05. The association of carbohydrate metabolism markers with perioperative characteristics was assessed using Spearman’s rank correlation. To identify factors independently associated with CABG in-hospital outcomes, we evaluated binary logistic regression (forward LR method) in two models: (1) presence of significant perioperative complications + extended hospital stay after surgery (>10 days); and (2) significant hospital complications. The model included factors such as glucose, type 2 diabetes mellitus, alternative markers of carbohydrate metabolism, overweightness or obesity, left atrial size, LV end-diastolic size, total perioperative parameters (duration of surgery, duration of CPB, number of shunts), preoperative heart rate at rest, and medical therapy, including hypoglycemic ones. Performance of carbohydrate metabolism preoperative parameters in discriminating the risk of the composite endpoint-1 development (significant perioperative complications + extended hospital stay after surgery) after CABG was evaluated through receiver operating characteristic curve analysis.

## 3. Results

A sample of 383 patients was divided into three groups depending on their glycemic status: Group 1—patients with DM 2 (*n* = 125), Group 2—patients with prediabetes (*n* = 67), and Group 3—patients without carbohydrate metabolism disorders (*n* = 191).

### 3.1. Initial Characteristics of Patients in the Groups of Diabetes Mellitus, Prediabetes, and Normoglycemia

Patients of the three groups did not differ in age, class of angina pectoris and heart failure, the prevalence of arterial hypertension, and the frequency of cardiovascular events in history (Table 1). There were significantly fewer men in the prediabetes and DM2 groups than in the normoglycemia group. The prediabetes and diabetes groups had a higher proportion of obese individuals compared with those without CMD (Table 1, Figure 2). Patients with diabetes had a higher incidence of carotid surgery and a lower incidence of smoking compared with the group without CMD (Table 1).

The median body mass index in patients with DM and prediabetes was significantly higher than the BMI of the group with normal glucose metabolism (Table 1). Patients did not differ in the main drug therapy before CABG, with the exception of hypoglycemic therapy, which was taken only by patients with DM (Table 1). In the DM group, 32.1% of patients took oral antihyperglycemic drugs before hospitalization and 15.2% received insulin. During hospitalization in preparation for CABG, 41.6% of patients with DM received insulin therapy (Table 1). Preoperative EuroSCORE II risk scores showed the lowest risk score among prediabetic individuals compared with the other two groups (Table 1). At the same time, medians of hospitalization of the DM and prediabetes groups were similar—13.0 days for the normoglycemia group—12.0 days (*p* < 0.001). 

Overweightness or obesity was very common in this cohort, occurring in 90.4% of diabetic patients, 85.1% of pre-diabetic patients, and 69.1% of the normoglycemic group. It is noteworthy that the percentage of overweightness, obesity, and long-term hospitalization among patients with prediabetes was no more favorable than in patients with DM and worse than in patients with normoglycemia (Figure 1). A similar trend, which did not reach statistical significance, was found for three-vessel disease, stem coronary artery disease, stenosis of the carotid arteries, and arteries of the lower extremities (Figure 1).

All groups were comparable in the frequency of cardiopulmonary bypass, combined surgery, duration of CPB, duration of aortic clamping, and other main characteristics of the surgery (Table 2). In the analysis of routine preoperative laboratory parameters, the median triglycerides were highest in the DM group compared with the other two groups; the rest of the indicators were comparable (Table 2).

The size of the left atrium was significantly larger in the DM group compared with the normoglycemia group (Table 2); there were no significant differences for other indicators. The previously noted trend that the median values of the groups with prediabetes are very close to the values of the DM group was also observed for echocardiographic parameters: linear and volumetric LV dimensions, left atrial dimensions, LV myocardial mass, LV myocardial mass index, and LV ejection fraction (Table 2).

### 3.2. Perioperative Dynamics of Carbohydrate Metabolism Markers in the Groups of Diabetes Mellitus, Prediabetes, and Normoglycemia

Figure 3 graphically reflects the fluctuations in the values of carbohydrate metabolism markers, and Table 2 shows their numerical values and critical significance levels *p,* both when comparing groups with each other and when comparing marker levels before and after surgery. Preoperative indicators of fructosamine and glucose naturally increased from the group without HMD to the DM group, with statistical significance when comparing each of the three groups with each other (Table 3, Figure 3), whereas on days 7–8 the differences remained significant only when comparing both groups with the DM group (Table 3). Moreover, the differences between the prediabetes and normoglycemia groups (1 and 2) disappeared due to the fact that on days 7–8 the median glucose values in the normoglycemia group increased and approached those of the prediabetes group (Table 3, Figure 3). At the same time, the level of fructosamine, which is an integral indicator of glucose for 3–4 weeks, decreased in the prediabetes group to the values of the normoglycemia group. At the same time, in all three groups there was a decrease in the median values of fructosamine when comparing preoperative and postoperative levels (Table 3). We can explain this by perioperative fasting and significantly lower carbohydrate intake up to 7–8 days after CABG and the fact that fructosamine was dynamic enough to reflect this condition. The median glucose also decreased on days 7–8 in groups 1 and 2. However, this was not the case in the normoglycemia group, where it even slightly increased in group 3 and became closer to the median of the prediabetes group, as indicated above.

Preoperative HbA1c levels naturally increased from the group without CMD to the DM group with statistical significance when comparing each of the three groups with each other, both before surgery and on days 7–8 (*p* < 0.001 in all cases) (Table 3, Figure 3). At the same time, there were no significant differences between preoperative and postoperative HbA1c values in any group. This is probably due to a slower change in glycated hemoglobin, which reflects glycemia in the 3 months prior to the study.

Attention is drawn to the following feature observed for almost all studied markers of carbohydrate metabolism except for HbA1c (glucose, fructosamine, and 1,5-anhydroglucitol), which is clearly visible in Figure 3: small quartile ranges before surgery, their proximity to the median in the prediabetes and normoglycemia groups before surgery, and a significant increase in the quartile ranges on days 7–8 after CABG. We can explain this by the fact that at rest and in the absence of diabetes mellitus, a high stability of carbohydrate metabolism is known, which was demonstrated by groups 1 and 2 (the diabetes group before the operation already had a large range of all markers). However, the operational stress caused a large amplitude of fluctuations in carbohydrate metabolism from hyperglycemia to hypoglycemia, and on the 7–8th day in all three groups we observe a high scatter of indicators between the 25th and 75th percentiles.

### 3.3. Correlation of Fructosamine and 1.5 Anhydroglucitol Levels before and after Surgery with Perioperative Characteristics of Patients

Furthermore, the correlation of fructosamine and 1,5-anhydroglucitol, determined before and on days 7–8 after CABG, with perioperative characteristics of patients was tested (Table 4). There was a direct correlation of the preoperative level of fructosamine with a variety of clinical characteristics: risk assessment according to EuroSCORE II, off-pump CABG, the number of cardioplegia, the duration of cardiopulmonary bypass and aortic clamping and the total duration of the operation, the number of shunts and distal anastomoses, body mass index, overweightness and obesity, the presence of type 2 diabetes mellitus, and the number of hospitalization days after CABG (Table 4). Laboratory parameters also directly correlated with the level of preoperative fructosamine–glucose before CABG and on days 7–8 after surgery, fructosamine after surgery, triglycerides, and fibrinogen level (Table 4). According to the result of echocardiography, there was a direct correlation of fructosamine with the size of the left atrium and an inverse one with an indicator of diastolic function (Vf). Fructosamine, determined on days 7–8, was associated with the presence of diabetes mellitus, body mass index, overweightness or obesity, intraoperative blood loss, off-pump CABG, LV posterior wall thickness and LV myocardial mass, diastolic indices (Vf), preoperative levels of triglycerides, and glucose and glycated hemoglobin determined upon admission and 7–8 days after CABG.

The preoperative level of 1.5 anhydroglucitol showed an inverse correlation with fasting glucose and fructosamine levels before surgery and on days 7–8 after CABG, the presence of type 2 diabetes mellitus, and intima media thickness and a direct correlation with end-diastolic volume. The postoperative level of 1.5 anhydroglucitol (days 7–8) was inversely correlated with combined operations, the presence of diabetes, operations on a beating heart, overweightness or obesity, and preoperative and postoperative levels of glucose and fructosamine.

### 3.4. Complications after CABG in the Groups of Diabetes Mellitus, Prediabetes, and Normoglycemia

For the majority of hospital complications, the trend remained—the prediabetes group was no more favorable in terms of the number of complications than DM and there were no statistically significant differences in the number of complications (Figure 4). The lethal outcome was in 2.6% of cases in the DM group and 1.8% in the normoglycemia group; there were no deaths in the hospital in the prediabetes group (*p* = 0.872). 

### 3.5. Factors Associated with the Development of Hospital Complications of CABG

We analyzed the incidence of combined endpoints. The first combined endpoint (presence of significant perioperative complications + extended hospital stay after surgery (>10 days)) was identified in 291 patients; 92 patients had no endpoint. Patients with a combined endpoint-1 were mostly women and were older; there was an association with DM, obesity, a higher preoperative EuroScore II risk, a longer CPB duration and the total duration of operations, the number of distal anastomosis, higher levels of glucose and fructosamine before surgery, an increase in the left atrium size, and the LV myocardium mass (Appendix A). In binary logistic regression analysis (Table 5), only patient age (*p* = 0.005) and fructosamine level (*p* = 0.022) were independently associated with the development of this composite endpoint-1. For this model, statistical significance was χ2(2) = 14.2, *p* = 0.001, the Nagelkerke R2 value was 0.11, and the model correctly classified 84.1% of cases. 

Patients with a composite endpoint-2 (significant hospital complications) were older and had higher LA dimensions, LV myocardial mass, presence of multifocal atherosclerosis, higher preoperative glucose levels, higher preoperative EuroScore risk, longer duration of CPB, and more shunts. In binary logistic regression analysis (Table 6), only patient age (*p* = 0.001), aortic occlusion time (*p* = 0.001), and number of shunts (*p* = 0.004) were independently associated with the development of this composite endpoint-2. For this model, statistical significance was χ2(2) = 15.9, *p* = 0.001, the Nagelkerke R2 value was 0.213, and the model correctly classified 77.2% of cases.

The association of carbohydrate metabolism preoperative parameters in discriminating the risk of the composite endpoint-1 development (significant perioperative complications + extended hospital stay after surgery) after CABG is presented in Figure 5. As shown in Appendix A, the areas under the curves were maximal for fructosamine before surgery (0.629, *p* = 0.001). The areas under the curves of other indicators were smaller, which indicated insufficient distinguishing ability.

## 4. Discussion

The present study shows that alternative markers of carbohydrate metabolism, fructosamine and 1,5-anhydroglucitol, are associated with different clinical characteristics in patients undergoing coronary bypass surgery. According to the regression analysis, the level of fructosamine was associated with one of the combined endpoints (significant hospital complications and long hospital stay after CABG).

So far, only a few studies have evaluated the association of fructosamine with the presence of perioperative complications in cardiac surgery. Thus, in a study by Kowalczuk-Wieteska et al. [10], they concluded that the levels of glycated hemoglobin and fructosamine equally determine the risk of perioperative complications in cardiac surgery patients. However, when analyzing the results of this study, it turned out that the preoperative level of these markers was higher in the group of patients without postoperative complications (although the authors did not reveal statistically significant differences). In contrast to this study, we found an association of increased fructosamine levels with the number of perioperative complications. In this regard, our results are consistent with the results of the prognostic value of fructosamine in patients undergoing primary total joint arthroplasty. This study showed that patients with fructosamine levels ≥292 mmol/L had a significantly higher risk of infectious complications (OR 6.2, *p* = 0.009), readmission (OR 3.0, *p* = 0.03), and reoperation (OR 3.4, *p* = 0.02). At the same time, there was no predictive value of HbA1c levels of ≥7% [18]. In a more recent multicenter study, similar results were obtained in patients with knee replacement surgery [19]. Apparently, the study of fructosamine as a prognostic marker is also deserved in cardiac surgery. 

Another marker of carbohydrate metabolism, 1,5-anhydroglucitol, has not yet been studied in cardiac surgery. In studies in patients with PCI, encouraging results were obtained in terms of its prognostic value. For example, Fujiwara et al. showed that low baseline values of 1,5-anhydroglucitol were associated with the development of adverse events in the prospective observation of patients after PCI. In a multivariate logistic analysis, low 1,5-AG values were independently associated with coronary revascularization or target vessel revascularization (*p* = 0.04 and *p* = 0.044, respectively) [20]. In another study by Takahashi et al. when observing patients for a year after PCI, it was noted that low and exacerbated levels of 1,5-anhydroglucitol were associated with cardiovascular events [14]. However, when analyzing the results obtained by the authors, it turned out that, according to the initial values, the groups with the presence and absence of subsequent complications of PCI did not differ; these differences appeared when assessing 1,5-anhydroglucitol in dynamics. In our opinion, it is still incorrect to interpret these data as a proven prognostic value of 1,5-anhydroglucitol in these patients. In addition, in patients after OCT-guided PCI, low 1,5-AG levels were not associated with the development of MACE, in contrast to the presence of diabetes mellitus [12]. In our study, we also did not reveal the prognostic value of 1,5-anhydroglucitol in patients with CABG, perhaps these data should be clarified in further studies in this direction.

What is the possible clinical significance of this study? A recent meta-analysis of 30 studies including 34,650 patients convincingly demonstrated that low glycated hemoglobin (<5.5%) before cardiac surgery is associated with a significant reduction in perioperative complications (such as death, acute kidney injury, neurological, and infectious complications) [5]. These data overcome the existing concerns among clinicians about tight perioperative glycemic control [21], and there is a proposal to achieve the maximum possible reduction in the level of glycated hemoglobin before surgery, not only in patients with DM but in general in all patients before cardiac surgery [22]. However, due to the longer period required to achieve the optimal values of glycated hemoglobin (up to 3 months), markers of carbohydrate metabolism such as fructosamine deserve attention. Alternative markers of carbohydrate metabolism (fructosamine and 1,5-anhydroglucitol) are easy to determine and may be useful for preoperative preparation and prediction of surgical outcomes. Until now, their use has been limited due to the lack of convincing data on the possible predictive value of these biomarkers. We hope that our study will initiate a more intensive study of this issue. 

### Study Limitation

The present study had several limitations that need to be taken into account when evaluating its results. First, we did not take into account the level of glycemic control, which could potentially affect the results of CABG. Second, in our study we did not investigate postoperative troponin levels, which could detect perioperative myocardial injury and allow a more accurate assessment of the contribution of fructosamine and 1,5-anhydroglucitol to the clinical outcomes of CABG. In addition, in the studied cohort of patients, we did not assess the level of glycated hemoglobin before surgery, which did not allow us to compare its prognostic value with the markers of carbohydrate metabolism that we studied. Also, to diagnose prediabetes in terms of fasting glucose and HbA1c, we used the WHO criteria, which are less strong than those of the American Diabetes Association [15,16,23]. 

## 5. Conclusions

This study demonstrated that in patients after CABG there was the significant decrease in the level of fructosamine compared with baseline, whereas the level of 1,5-anhydroglucitol did not change. Preoperative fructosamine levels were one of the independent predictors of the combined endpoint. The prognostic value of preoperative assessment of alternative markers of carbohydrate metabolism in cardiac surgery deserves further study.

## Figures and Tables

**Figure 1 diagnostics-13-00969-f001:**
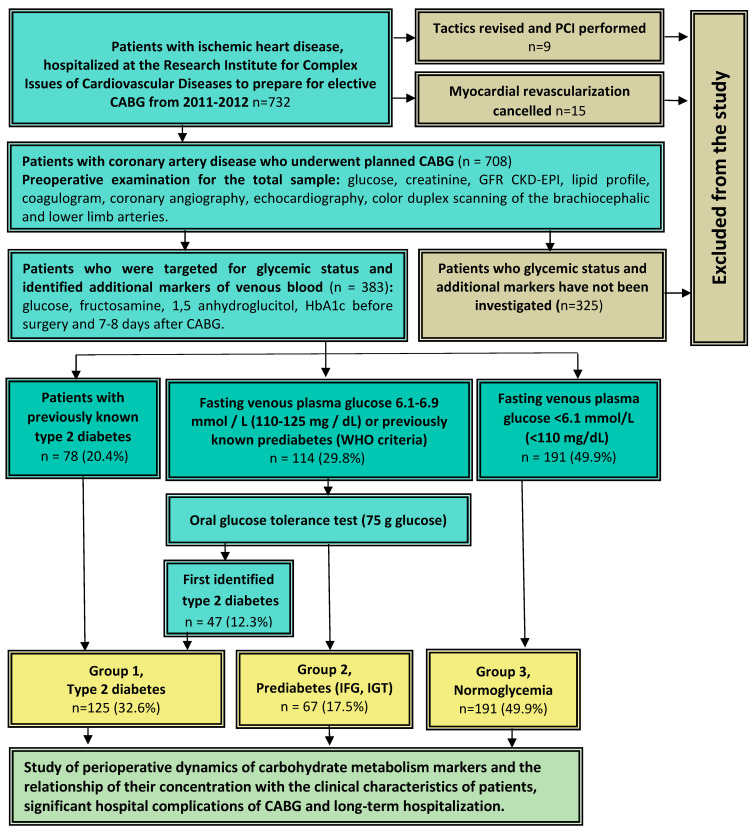
Study design. Notes: CABG—coronary artery bypass grafting, IVR—disorders of carbohydrate metabolism, IFG—impaired fasting glycemia, IGT—impaired glucose tolerance, GFR—glomerular filtration rate, CKD-EPI—Chronic Kidney Disease Epidemiology Collaboration.

**Figure 2 diagnostics-13-00969-f002:**
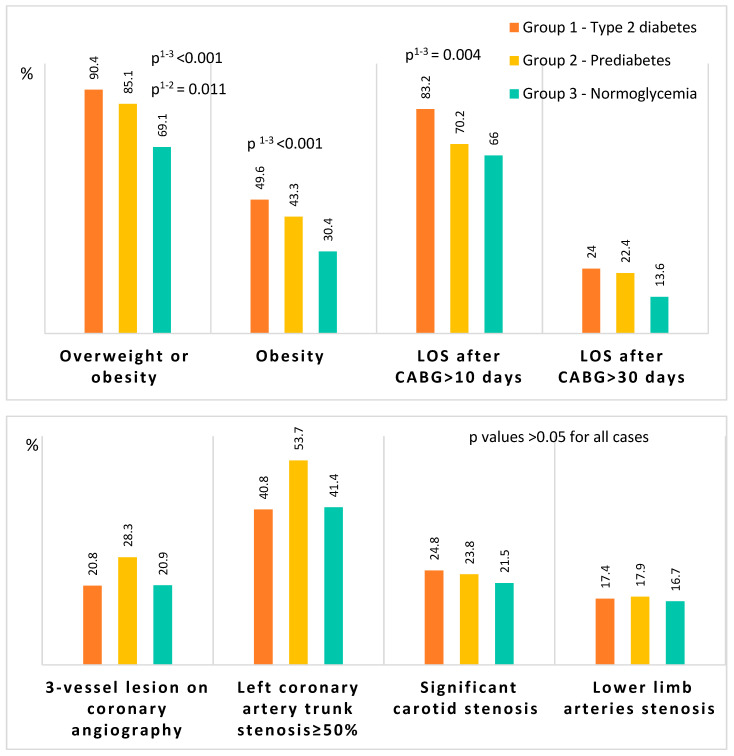
Overweightness, duration of hospitalization stay, and data from instrumental examinations of the coronary and non-coronary arteries. Notes: BMI—body mass index, CABG—coronary artery bypass grafting, LOS—length of hospital stay. In other cases, *p* values are greater than the critical significance level.

**Figure 3 diagnostics-13-00969-f003:**
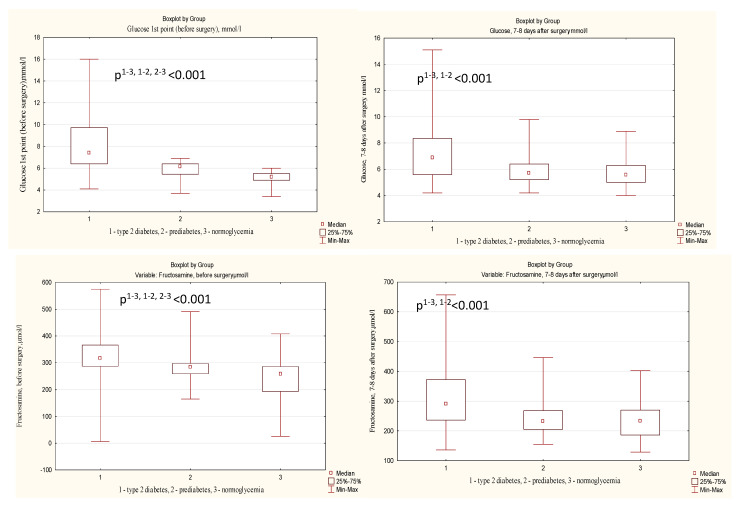
Increase in interquartile ranges of markers of carbohydrate metabolism in groups after CABG in comparison with preoperative values.

**Figure 4 diagnostics-13-00969-f004:**
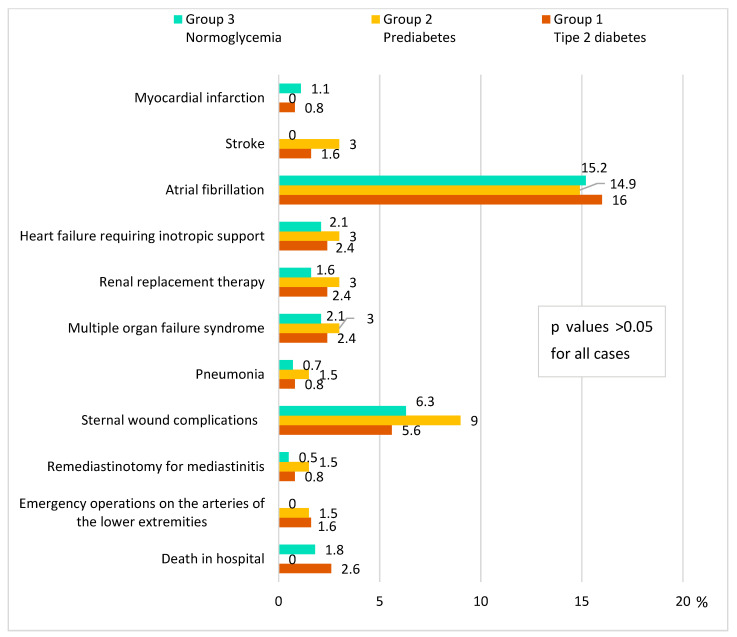
Hospital complications of CABG in groups.

**Figure 5 diagnostics-13-00969-f005:**
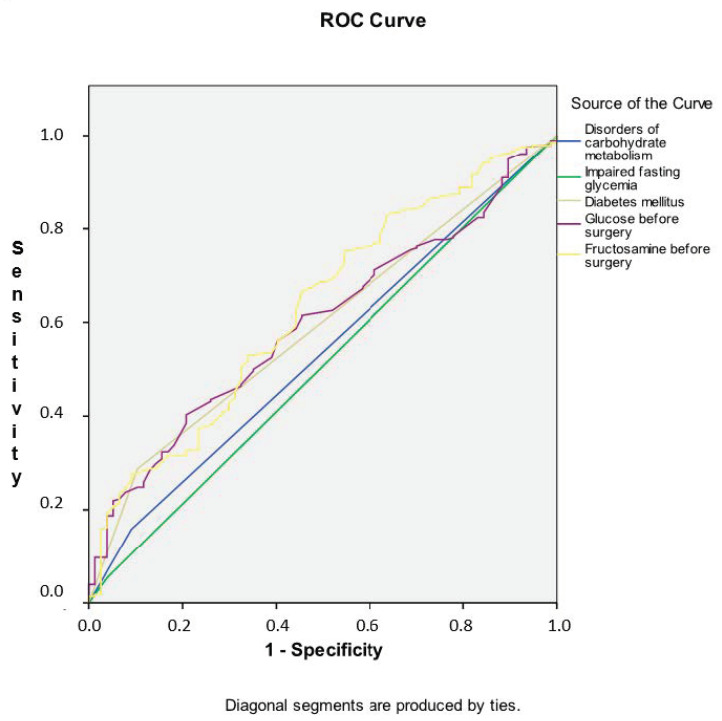
Receiver operating characteristic curve analysis. Performance of carbohydrate metabolism preoperative parameters in discriminating the risk of the composite endpoint-1 development (significant perioperative complications + extended hospital stay after surgery) after CABG. Notes: ROC, receiver operating characteristic systolic excursion.

**Table 1 diagnostics-13-00969-t001:** Anamnestic and clinical characteristics of patients and preoperative medicine therapy.

Parameter	Group 1 Type 2 Diabetes *n* = 125	Group 2 Prediabetes *n* = 67	Group 3 Normoglycemia *n* = 191	*p*
Men (*n*, %)	79 (63.2)	51 (76.1)	154 (80.6)	<0.001 *
Age, years (Me [LQ; UQ])	59.0 [54.5; 63.0]	58.0 [54.5; 63.0]	59.0 [54.0; 65.0]	0.493
BMI, kg/m ^2^ (Me [LQ; UQ])	29.8 [27.2; 32.6]	29.3 [26.8; 32.3]	27.0 [24.2; 30.9]	<0.001 *
Overweight or obesity (BMI ≥25 kg/m^2^, *n*, %)	113 (90.4)	57 (85.1)	132 (69.1)	<0.001 * 0.011 #
Obesity (BMI ≥30 kg/m^2^) (*n*, %)	62 (49.6)	29 (43.3)	58 (30.4)	<0.001 *
Arterial hypertension (*n*, %)	117 (93.6)	61 (91.0)	165 (86.4)	0.111
Angina class III-IV (*n*, %)	48 (38.4)	26 (38.8)	71 (37.1)	0.802
Heart failure class NYHA III–IV (*n*, %)	41 (32.8)	19 (28.3)	52 (27.2)	0.803
Ventricular arrhythmias (*n*, %)	21 (16.8)	9 (13.4)	26 (13.6)	0.751
Supraventricular arrhythmias (*n*, %)	14 (11.2)	6 (8.9)	13 (6.8)	0.100
Intermittent claudication (*n*, %)	18 (14.4)	8 (13.4)	27 (14.1)	0.445
Smoking/smoking (*n*, %)	29 (23.2)	20 (29.9)	79 (41.4)	<0.001 *
History of myocardial infarction (*n*, %)	81 (64.8)	41 (61.2)	120 (62.8)	0.665
History of stroke (*n*, %)	9 (7.2)	3 (4.5)	15 (7.8)	0.647
Previous PCI (*n*, %)	16 (8.4)	6 (7.5)	16 (8.3)	0.342
Previous CABG (*n*, %)	2 (1.6)	0 (0)	2 (1.1)	0.582
Surgery on the carotid arteries (*n*, %)	10 (6.4)	2 (3.0)	1 (0.5)	0.002 *
Intervention on the arteries lower limbs or amputation (*n*, %)	2 (1.6)	0 (0)	1 (0.5)	0.413
EuroSCORE II, % (Me [LQ; UQ])	2.1 [1.3; 3.5]	1.5 [0.9; 2.3]	1.7 [1.2; 2.6]	0.008 # 0.023 *
EuroSCORE II, points (Me [LQ; UQ])	3 [2; 5]	3 [1; 3]	3 [2; 4]	0.006 * 0.011 #
Length of hospital stay after CABG, days (Me [LQ; UQ])	13.0 [11.0; 17.0]	13.0 [9.0; 15.0]	12.0 [10.0; 14.0]	<0.001 *
Preoperative medicine therapy (*n*, %)				
β-Blockers	124 (99.2)	66 (98.5)	186 (97.4)	0.621
Angiotensin converting enzyme inhibitors	109 (87.2)	59 (88.1)	169 (88.5)	0.731
Angiotensin 2 receptor antagonists	9 (7.2)	2 (3.0)	5 (4.5)	0.293
Mineralocorticoid receptor antagonists	22 (17.6)	12 (17.9)	32 (16.8)	0.765
Thiazide-like diuretics	16 (12.9)	5 (7.4)	18 (9.4)	0.182
Loop diuretics	10 (8.0)	4 (6.0)	11 (5.8)	0.306
Calcium channel blockers	97 (77.6)	42 (62.6)	111 (58.1)	0.043
HMG-CoA reductase inhibitors (statins)	96 (76.8)	49 (73.1)	143 (74.9)	0.923
Only oral hypoglycemic drugs	41 (21.3)	-	-	-
Metformin	72 (37.6)	-	-	-
Sulfonylurea	2 8 (22.4)	-	-	-
DPP-4 inhibitors or GLP-1 receptor agonists	5 (4.0)	-	-	-
Insulin therapy before hospitalizations	19 (15.2)	-	-	-
Insulin therapy in time hospitalizations	52 (41.6)	-	-	-

**Notes**: Me [LQ; UQ]—median with upper and lower quartile, BMI—body mass index, NYHA—New York Heart Association, PCI—percutaneous coronary intervention, CABG—coronary artery bypass grafting, EuroSCORE II—European Cardiovascular Risk Score, HMG-CoA—hydroxymethylglutaryl-coenzyme A, DPP-4—dipeptidyl peptidase 4, GLP-1—glucagon-like peptide 1. *—significant differences in pairwise comparison of groups 1 and 3, #—significant differences in pairwise comparison of groups 2 and 3.

**Table 2 diagnostics-13-00969-t002:** Characteristics of the operation and routine laboratory parameters.

Parameter	Group 1 Type 2 Diabetes *n* = 125	Group 2 Prediabetes *n* = 67	Group 3 Normoglycemia *n* = 191	*p*
Characteristic of CABG				
Use of cardiopulmonary bypass (*n*, %)	118 (94.4)	59 (88.1)	169 (88.5)	0.179
Isolated CABG (*n*, %)	111 (88.8)	63 (94.0)	178 (93.2)	0.128
Combined operations (*n*, %)	14 (11.2)	4 (6.0)	11 (5.8)	0.128
Carotid endarterectomy (*n*, %)	2 (1.6)	3 (4.5)	3 (1.6)	0.322
Ventriculoplasty (*n*, %)	8 (6.4)	1 (1.4)	5 (2.6)	0.024
Radio frequency ablation (*n*, %)	7 (5.6)	2 (3.0)	4 (2.1)	0.238
Mitral valve (*n*, %)	0 (0)	0 (0)	1 (0.5)	0.604
Aortic valve (*n*, %)	0 (0)	1 (1.5)	2 (1.1)	0.451
Cardiopulmonary bypass duration, minutes (Me [ LQ; UQ])	100.0 [81.0; 118.0]	92.5 [75.0; 109.0]	95.0 [78.0; 109.0]	0.115
Aortic clamp time, minutes (Me [LQ; UQ])	66.5 [51.5; 78.0]	58.0 [49.0; 67.0]	60 [49.0; 72.0]	0.064
Total operation time, minutes (Me [LQ; UQ])	246.0 [204.0; 300.0]	240.0 [201.0; 270.0]	240.0 [198.0; 264.0]	0.114
Intraoperative blood loss, ml (Me [LQ; UQ])	500.0 [500.0; 600.0]	500.0 [500.0; 500.0]	500.0 [500.0; 500.0]	0.117
Number of shunts (Me [LQ; UQ])	3.0 [2.0; 3.0]	2.0 [2.0; 3.0]	2.0 [2.0; 3.0]	0.089
Complete revascularization (*n*, %)	1 17 (93.6)	61 (91.0)	172 (90.1)	0.534
Preoperative laboratory fasting blood values (Me [LQ; UQ])				
Total cholesterol, mmol/L/	5.2 [4.2; 6.2]	4.6 [3.8; 5.8]	5.0 [4.2; 6.0]	0.079
HDL cholesterol, mmol/L	0.9 [0.8; 1.1]	0.96 [ 0.8; 1.1]	1.0 [0.9; 1.2]	0.095
LDL cholesterol mmol/L	3.0 [2.3; 4.1]	2.5 [ 6.3; 2.1]	2.9 [2.3; 3.7]	0.150
Triglycerides, mmol/L	2.1 [1.5; 2.7]	1.9 [1.2; 2.2]	1.6 [1.2; 2.2]	<0.001 *
Creatinine, µmol/L	83.0 [68.0; 95.0]	85.0 [71.0; 104.0]	83.0 [74.0; 106.0]	0.098
GFR CKD—EPI, mL/min/1.73 m^2^	82.5 [69.2; 99.7]	80.3 [62, 3; 100.0]	82.4 [66.3; 103.5]	0.297
Fibrinogen, g/L	4.8 [3.8; 6.0]	4.4 [3.5; 5.3]	4.4 [3.5; 5.6]	0.187
Preoperative echocardiography (Me [LQ; UQ])				
End-diastolic LV volume (mL)	160.0 [136.0;194.5]	160.0 [140.0;185.0]	154.0 [132.5; 185.0]	0.123
End-diastolic LV dimension (cm)	5.6 [5.3; 6.2]	5.7 [5.3; 6.0]	5.5 [5.1; 6.0]	0.324
End-systolic LV volume (mL)	67.5 [51.0; 104.0]	66.0 [51.0; 88.0]	59.5 [44.0; 91.0]	0.034
End-systolic LV dimension (cm)	4.0 [3.5; 4.9]	3.9 [3.5; 4.4]	3.7 [3.3; 4.5]	0.019
Left atrium (cm)	4.3 [4.0; 4.6]	4.3 [3.9; 4.5]	4.2 [3.8; 4.4]	0.014 *
LV ejection fraction (%)	58.0 [48.0; 64.0]	60.0 [52.0; 64.0]	62.0 [52.0; 65.0]	0.107
LV myocardial mass by Deveraux and Reichek, g	312.3 [258.5; 372.0]	292.5 [255.8; 353.9]	292.5 [241.1; 370.0]	0.029
LV myocardial mass index, g/m^2^	165.0 [140.0; 188.0]	153.4 [129.9; 188.0]	155.0 [126.2; 188.1]	0.119
Stroke volume (mL)	90.2 [80.0; 103.0]	90.1 [79.0; 102.0]	89.0 [76.0; 103.0]	0.279

Notes: CMD—carbohydrate metabolism disorders, Me [LQ; UQ]—median with upper and lower quartile, HDL—high-density lipoproteins, LDL—low-density lipoproteins, GFR—glomerular filtration rate, CKD-EPI—Chronic Kidney Disease Epidemiology Collaboration, FV—flow velocity, E/FV—the ratio of the peak of the early transmitral flow to the early diastolic flow velocity in the LV cavity. *—significant differences in pairwise comparison of groups 1 and 3.

**Table 3 diagnostics-13-00969-t003:** The values of fructosamine, 1,5-anhydroglucitol, and glucose in groups and their perioperative dynamics (*n* = 383).

Parameter (Me [LQ; UQ])	Group 1 Type 2 Diabetes *n* = 125	Group 2 Prediabetes *n* = 67	Group 3 Normoglycemia *n* = 191	*p*
Glucose 1st point (before surgery), mmol/L	7.6 [6.4; 9.9]	6.2 [6.0; 6.5]	5.2 [4.9; 5.5]	<0.001 * # $
Glucose 2nd point (7–8 days after surgery), mmol/L	6,9 [5.6; 8.35]	5.7 [5.2; 6.4]	5.6 [5.0; 6.3]	<0,001 * $
p1st–2nd point (for glucose)	0.003	0.011	<0.001	
Fructosamine 1st point (before surgery), µmol/L	317 [287;366]	284 [259; 297]	245 [194; 285]	<0.001 * # $
Fructosamine 2nd point (7–8 days after surgery), µmol/L	291 [236.5; 372.5]	233 [205; 268]	234 [186; 270]	<0.001 * $
p1st–2nd point (for FA)	0.030	0.001	0.038	
1.5 Anhydroglucitol 1st point (before surgery), mcg/mL	17.72 [14.2; 21.8]	23.37 [19.99; 26.08]	23.64 [19.8; 28.9]	<0.001 * $
1.5 Anhydroglucitol 2nd point (7–8 days after surgery), mcg/mL	17.44 [14.95; 20.62]	21.34 [15.80; 26.25]	20.93 [17.64; 24.22]	<0.001 *
p1st–2nd point (for 1.5 AG)	0.247	0.674	0.092	
HbA1c 1st point (before surgery), %	7.3 [5.9; 8.2]	5.4 [5.3; 6.0]	5.1 [5.0; 5.6]	<0.001 * # $
HbA1c 2nd point (7–8 days after surgery), %	7.2 [5.8; 8.5]	5.4 [5.1; 6.0]	5.1 [5.1; 5.8]	<0.001 * # $
p1st–2nd point (for HbA1c)	0.181	0.601	0.100	

Notes: FA—fructosamine, 1.5 AG—1,5-anhydroglucitol, HbA1c—glycated hemoglobin A1c, *—significant differences in pairwise comparison of groups 1 and 3, #—significant differences in pairwise comparison of groups 2 and 3, $—significant differences in pairwise comparison of groups 1 and 2.

**Table 4 diagnostics-13-00969-t004:** Correlation of fructosamine and 1.5-anhydroglucitol with clinical characteristics of patients.

Correlates	Spearman-R	*p*-Value
Fructosamine 1st point (before surgery)		
Fructosamine 1st point and Type 2 diabetes	0.505	<0.001
Fructosamine 1st point and EuroSCORE (points)	0.196	<0.001
Fructosamine 1st point and EuroSCORE (%)	0.152	0.002
Fructosamine 1st point and Off-pump	0.113	0.033
Fructosamine 1st point and Multiplicity of cardioplegia	0.197	<0.001
Fructosamine 1st point and Cardiopulmonary bypass duration	0.204	<0.001
Fructosamine 1st point and Aortic clamp time	0.177	<0.001
Fructosamine 1st point and Number of distal anastomoses	0.129	0.016
Fructosamine 1st point and Number of shunts	0.124	0.012
Fructosamine 1st point and Total operation duration	0.120	0.027
Fructosamine 1st point and Body mass index	0.176	<0.001
Fructosamine 1st point and Overweight or obesity	0.175	<0.001
Fructosamine 1st point and Left atrium	0.117	0.028
Fructosamine 1st point and Flow velocity	−0.193	0.030
Fructosamine 1st point and LOS after CABG	0.206	<0.001
Fructosamine 1st point and Triglycerides	0.215	<0.001
Fructosamine 1st point and Fibrinogen	0.155	0.002
Fructosamine 1st point and Glucose 1st point	0.480	<0.001
Fructosamine 1st point and Glucose 2nd point	0.249	<0.001
Fructosamine 1st point and HbA1c 1st point	0.379	<0.001
Fructosamine 1st point and HbA1c 2nd point	0.280	<0.001
Fructosamine 1st point and Fructosamine 2nd point	0.265	<0.001
Fructosamine 2nd point (7–8 days after surgery)		
Fructosamine 2nd point and Type 2 diabetes	0.351	<0.001
Fructosamine 2nd point and Body mass index	0.134	0.019
Fructosamine 2nd point and Overweight or obesity	0.145	0.011
Fructosamine 2nd point and Intraoperative blood loss	0.120	0.037
Fructosamine 2nd point and Off-pump	−0.128	0.032
Fructosamine 2nd point and Posterior wall of the LV	0.135	0.018
Fructosamine 2nd point and LV myocardial mass by Deveraux and Reichek	0.121	0.033
Fructosamine 2nd point and Flow velosity	−0.184	0.049
Fructosamine 2nd point and Triglycerides	0.172	0.005
Fructosamine 2nd point and Glucose 1st point	0.324	<0.001
Fructosamine 2nd point and Glucose 2nd point	0.935	<0.001
Fructosamine 2nd point and HbA1c 1st point	0.268	<0.001
Fructosamine 2nd point and HbA1c 2nd point	0.172	<0.001
1.5 Anhydroglucitol 1st point (before surgery)		
1.5 Anhydroglucitol 1st point and Type 2 diabetes	−0.458	<0.001
1.5 Anhydroglucitol 1st point and Intima media thickness	−0.194	0.016
1.5 Anhydroglucitol 1st point and LV end-diastolic volume	0.183	0.020
1.5 Anhydroglucitol 1st point and Glucose 1st point	−0.527	<0.001
1.5 Anhydroglucitol 1st point and Glucose 2nd point	−0.309	<0.001
1.5 Anhydroglucitol 1st point and 1.5 anhydroglucitol 2nd point	0.586	<0.001
1.5 Anhydroglucitol 1st point and Fructosamine 1st point	−0.198	0.012
1.5 Anhydroglucitol 1st point and Fructosamine 2nd point	−0.302	<0.001
1.5 Anhydroglucitol 2nd point (7–8 days after surgery)		
1.5 Anhydroglucitol 2nd point and Type 2 diabetes	−0.387	0.004
1.5 Anhydroglucitol 2nd point and Overweight or obesity	−0.277	0.047
1.5 Anhydroglucitol 2nd point and Off-pump	−0.346	0.011
1.5 Anhydroglucitol 2nd point and Combined operations	−0.273	0.048
1.5 Anhydroglucitol 2nd point and Glucose 1st point	−0.433	0.001
1.5 Anhydroglucitol 2nd point and Glucose 2nd point	−0.349	0.012
1.5 Anhydroglucitol 2nd point and Fructosamine 1st point	−0.414	0.003
1.5 Anhydroglucitol 2nd point and Fructosamine 2nd point	−0.336	0.016

Notes: LV—left ventricle, IMT—intima media thickness, E/FV—ratio of early transmitral flow peak to early diastolic flow velocity in the LV cavity, EuroSCORE—European System for Cardiac operational risk Evaluation, BMI—body mass index, CABG—coronary artery bypass grafting, LOS—length of hospital stay. All instrumental parameters—preoperative, all laboratory parameters—fasting, determined before surgery unless otherwise indicated.

**Table 5 diagnostics-13-00969-t005:** Factors associated with the development of this composite endpoint-1 (significant perioperative complications + extended hospital stay after surgery) after CABG (binary logistic regression analysis, forward LR method).

		B	S.E.	Wald	df	Sig.	Exp(B)
Step 1 ^a^	Age	0.074	0.026	8.077	1	0.004	1.077
	Constant	−2.677	1.519	3.104	1	0.078	0.069
Step 2 ^b^	Age	0.074	0.026	7.946	1	0.005	1.076
	Fructosamine	0.007	0.003	5.226	1	0.022	1.007
	Constant	−4.561	1.744	6.840	1	0.009	0.010
a. Variable(s) entered on step 1: Age							
b. Variable(s) entered on step 2: Fructosamine							

**Table 6 diagnostics-13-00969-t006:** Factors associated with the development of this composite endpoint-2 (significant perioperative complications) after CABG (binary logistic regression analysis, forward LR method).

		B	S.E.	Wald	df	Sig.	Exp(B)
Step 1 ^a^	Age	0.096	0.025	15.230	1	0.000	1.101
	Constant	−7.052	1.559	20.450	1	0.000	0.001
Step 2 ^b^	Age	0.112	0.027	17.889	1	0.000	1.119
	Aortic occlusion time	0.026	0.008	9.725	1	0.002	1.026
	Constant	−9.760	1.911	26.086	1	0.000	0.000
Step 3 ^c^	Age	0.137	0.029	21.972	1	0.000	1.147
	Aortic occlusion time	0.043	0.011	15.942	1	0.000	1.044
	Shunt count	−0.913	0.314	8.470	1	0.004	0.401
	Constant	−10.029	1.975	25.774	1	0.000	0.000
a. Variable(s) entered on step 1: Age.							
b. Variable(s) entered on step 2: Aortic occlusion time							
c. Variable(s) entered on step 3: Shunt count							

## Data Availability

The datasets used and/or analyzed during the current study available from the corresponding author on reasonable request.

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
