# Peer review of "Association of Alternative Markers of Carbohydrate Metabolism (Fructosamine and 1,5-Anhydroglucitol) with Perioperative Characteristics and In-Hospital Complications of Coronary Artery Bypass Grafting in Patients with Type 2 Diabetes Mellitus, Prediabetes, and Normoglycemia"

_diagnostics, 2023, doi:10.3390/diagnostics13050969_

Round 1

Reviewer 1 Report (Previous Reviewer 1)

I would like to congratulate the authors on the selection of this novel and the interesting aspect of cardiology. The authors have provided good evidence to support their conclusion with well constructed and meticulously written manuscript. 

However, I would like to bring attention to the following points

1. Please elaborate statistical section on the methodology 

2. In the results section provided results in graphical format can be helpful for readers 

3. In the limitations section more succinct presentation is needed. 

Author Response

I would like to congratulate the authors on the selection of this novel and the interesting aspect of cardiology. The authors have provided good evidence to support their conclusion with well-constructed and meticulously written manuscript. 

We would like to thank the reviewer for the kind review of our article. Your comments helped us correct the article and improve it. We also additionally showed our manuscript to a native English speaker.

However, I would like to bring attention to the following points

  1. Please elaborate statistical section on the methodology 

We've added ROC Analysis usage data to the Statistical Analysis section

  1. In the results section provided results in graphical format can be helpful for readers 

Figures 2-4 present some of the received results in graphical format.

  1. In the limitations section more succinct presentation is needed. 

We have adjusted the study limitation section by shortening it.

Reviewer 2 Report (New Reviewer)

This article examines the postoperative effects of fructosamine and 1,5 anhydroglucitol on CABG patients. The paper focuses on fructosamine and 1,5 anhydroglucitol and evaluates them in detail, but the following points need to be considered.

#1 Fructosamine and 1,5 anhydroglucitol are examined, but are the results different from those examined in the presence of A1C and especially diabetes mellitus? It seems to me that you simply take it as stating that diabetics have more complications. The authors should compare how fructosamine is superior to diabetes and A1C regarding the influence on endopoints, using ROC analysis, etc.

#2 The composite endpoint of this study is hospitalization for more than 10 days after surgery in addition to major vascular events. Is this endpoint a general indicator? The authors should indicate the reason for using this endopoint. Furthermore, the authors should consider the effect of this endopoint on the commonly used major cardiovascular events.

Author Response

This article examines the postoperative effects of fructosamine and 1,5 anhydroglucitol on CABG patients. The paper focuses on fructosamine and 1,5 anhydroglucitol and evaluates them in detail, but the following points need to be considered.

We would like to thank the reviewer for the kind review of our article. Your comments helped us correct the article and improve it.

#1 Fructosamine and 1,5 anhydroglucitol are examined, but are the results different from those examined in the presence of A1C and especially diabetes mellitus? It seems to me that you simply take it as stating that diabetics have more complications. The authors should compare how fructosamine is superior to diabetes and A1C regarding the influence on endopoints, using ROC analysis, etc.

Thank you for your suggestion. We additionally conducted a ROC analysis and included its results in the text of the manuscript. The association of carbohydrate metabolism preoperative parameters in discriminating the risk of the composite endpoint-1 development (significant perioperative complications + extended hospital stay after surgery) after CABG is presented in Figure 5 and Suppl. Table S2. Based on the results of the ROC analysis, the results of the logistic regression analysis on the independent effect of fructosamine on the development of composite endpoint-1 were confirmed. Of all the studied indicators of carbohydrate metabolism, the largest area under the curve turned out to be for fructosamine.

#2 The composite endpoint of this study is hospitalization for more than 10 days after surgery in addition to major vascular events. Is this endpoint a general indicator? The authors should indicate the reason for using this endopoint. Furthermore, the authors should consider the effect of this endopoint on the commonly used major cardiovascular events.

Because MACE (stroke, myocardial infarction, and CV death) were rare in the groups, logistic regression did not reveal any patterns. When all significant complications were taken into account, the number of endpoints was higher, which made it possible to establish predictors of their occurrence (age, time of aortic clamping, number of shunts applied). At the same time, none of the markers of carbohydrate metabolism (both traditional and alternative) were associated with an unfavorable outcome. The next step was to study such a point, which would include both the duration of hospitalization and significant complications. In this case, fructosamine, along with age, was a predictor of this event (hospitalization >10 days or significant complication). This combined endpoint is not generally accepted, which is one of the limitations of our study, but it is acceptable to use such a combined endpoint.

Round 2

Reviewer 2 Report (New Reviewer)

I have no further comments regarding the revised manuscript.

This manuscript is a resubmission of an earlier submission. The following is a list of the peer review reports and author responses from that submission.

Round 1

Reviewer 1 Report

I would like to congratulate the authors on this novel manuscript. It does appear to be a promising aspect for patients after CABG. 

However, I would have following suggestions for the authors

1. Provide a more in-depth introduction with appropriate citations

2. Please provide detailed descriptions of statistical methods

3. Limitations paragraph needs to be eloborated. 

Author Response

I would like to congratulate the authors on this novel manuscript. It does appear to be a promising aspect for patients after CABG. 

Reply: We would like to thank the reviewer for the work done in reviewing our manuscript and for the favorable feedback. The reviewer's comments helped us improve our manuscript.

However, I would have following suggestions for the authors

  1. Provide a more in-depth introduction with appropriate citations

Reply: Apparently, the distinguished reviewer would like us to present in more detail the pathophysiological and biochemical aspects of the use of alternative markers of carbohydrate metabolism in patients. However, our study is predominantly clinical in nature, so we considered it possible only to present a detailed clinical rationale for the study of alternative markers of carbohydrate metabolism in patients with CABG.

  1. Please provide detailed descriptions of statistical methods

Reply: We have corrected the text. We added: “The association of carbohydrate metabolism markers with perioperative characteristics was assessed using Spearman's rank correlation. To identify factors independently associated with CABG in-hospital outcomes, we evaluated binary logistic regression (Forward LR method) in two models: 1) presence of significant perioperative complications + extended hospital stay after surgery (> 10 days); 2) significant hospital complications. The model included factors such as glucose, type 2 diabetes mellitus, alternative markers of carbohydrate metabolism, overweight or obesity, left atrial size, LV end-diastolic size, total perioperative parameters (duration of surgery, duration of CPB, number of shunts), preoperative heart rate at rest, medical therapy, including hypoglycemic.

  1. Limitations paragraph needs to be elaborated. 

Reply: We have corrected the text of the section Study limitation. We added: “Also, to diagnose prediabetes in terms of fasting glucose and HbA1c, we used the WHO criteria, which are less strong than those of the American Diabetes Association [15, 16, 23].”

Author Response

Although this study presents an interesting investigation of carbohydrate metabolism in patients with CABG, the following concerns need to be addressed before publications.

Reply: We would like to thank the reviewer for the work done in reviewing our manuscript and for the favorable feedback. The reviewer's comments helped us improve our manuscript.

  1. In the title, there is no need to stick out fructosamine and 1,5 anhydroglucitol, since the authors also investigate other carbohydrate metabolism markers.

Reply: Dear reviewer, we consider it reasonable to leave these markers in the title, since the rest of the indicators were previously studied and are the standard for controlling carbohydrate metabolism, including before surgical operations, while fructosamine and 1.5 anhydroglucitol were studied for the first time in CABG.

  1. In the abstract, the reviewer suggested to add a background that introduce the role of carbohydrate metabolism in determining hospital complications of CABG to increase readers' interest in reading.

Reply: We made a correction in the abstract text, we added: “Patients with type 2 diabetes make up 25 to 40% of those referred for coronary bypass surgery, and the impact of diabetes on the results of the operation is studied in various aspects. To assess the state of carbohydrate metabolism before any surgical interventions, including CABG, daily glycemic control and the determination of glycated hemoglobin (HbA1c) are recommended. Glycated hemoglobin reflects the glucose concentration for the 3 months prior to the measurement, but alternative markers that reflect glycemic fluctuations over a shorter period of time may be useful in preoperative preparation.”

  1. In the abstract, CABG represents “coronary artery bypass surgery”; in the introduction, it means “coronary artery bypass grafting”. This makes it very difficult for readers to understand.

Reply: We have corrected the text.

  1. “DM” conditions is an important reason for initiating the study. The authors should added this information in the title, and the background of abstract.

Reply: We have corrected the title. New version of the title: “Association of alternative markers of carbohydrate metabolism (fructosamine and 1,5 anhydroglucitol) with perioperative characteristics and in-hospital complications of coronary artery bypass grafting in patients with type 2 diabetes mellitus, prediabetes, and normoglycemia”

  1. Some uses of language need editing, for example, Lines 57-60, Lines 65-67.

Reply: Unfortunately, the manuscript for revision does not contain line numbers, so we were unable to identify the portion of the manuscript in need of revision.

  1. Figure 1 is missing.

Reply: Figure 1 added.

  1. Line 172, please make sure that Figure 1 demonstrate the obese data, not Figure 2? Figure labels are confusing.

Reply: Yes, this is our mistake, we are talking about Figure 2, we have made corrections. We also made the notes in figure 2 more understandable.

  1. Table 4, Typing mistakes, “<0,001” should be “< 0.001”.

Reply: We have corrected the text

  1. Conclusion needs to be rewritten. The sentence “in patients after CABG there was the significant decrease in the level of fructosamine compared with baseline, while the level of 1,5-anhydroglucitol did not change. ” demonstrated a result, not a conclusion.

Reply: We have corrected the text. New conclusion: “This study demonstrated that in patients after CABG there was the significant decrease in the level of fructosamine compared with baseline, while the level of 1,5-anhydroglucitol did not change. Preoperative fructosamine levels were one of the independent predictors of the combined endpoint. The prognostic value of preoperative assessment of alternative markers of carbohydrate metabolism in cardiac surgery deserves further study”.

Reviewer 3 Report

The topic is quite interesting, although execution could be somewhat better, starting with major styling problems and excessive amount of info.

Abstract should include numerical data if the main results

The name of test is Kruskal-Wallis, not Kruskell-Wallace 

The authors should delinate how divison to groups was performed, i.e. Which criteria were used to designate prediabetes and normoglycemia. Specifically, the authors designated that HbA1c <6.0 is considered normal, although according to contemporary ADA guidelines this is already cinsidered prediabetes. In this regard, it can be realized from IQR that some HbA1c levels are above 5,6 in the normoglycemic group. This is an important point that should be carefully addressed, given that most comparisons rely on this disbiguation.

Figure 2. Minor point - Something is wrong with the footnote (part of it is at the end of the page) instead after figure

Figure 2. Measurement of data dispersion should be included in the figure, alongside p values (please apply this to all other figures as well)

I wonder which factors were considered in regression analysis.

As groups differ very much by sex, BMI, smoking status… I wonder what do results are after adjusting for these.

Hypoglycemic therapy (including sglt2i) is a conditio sine qua non for this analysis, in my opinion for obvious reasons.

Overall, given that most of the reported correlations concerning primary endpoint, as well as logistic regression imply only weak correlation (barely even that), alongside the fact that not all factors were considered, I would be very careful in concluding based on this data. Hence, discussion section must clearly adresss the weakness of these results, and lack of added clinical benefit. My opinion that the idea for the study is valid, yet it was not well-executed. The observed increase in patients with complications is probably a marker of higher cortisol response in this group.

Author Response

The topic is quite interesting, although execution could be somewhat better, starting with major styling problems and excessive amount of info.

Reply: We would like to thank the reviewer for the work done in reviewing our manuscript and for the favorable feedback. The reviewer's comments helped us improve our manuscript.

Abstract should include numerical data if the main results

Reply: We made a correction in the abstract text

The name of test is Kruskal-Wallis, not Kruskell-Wallace 

Reply: We have corrected the text.

The authors should delinate how divison to groups was performed, i.e. Which criteria were used to designate prediabetes and normoglycemia. Specifically, the authors designated that HbA1c <6.0 is considered normal, although according to contemporary ADA guidelines this is already cinsidered prediabetes. In this regard, it can be realized from IQR that some HbA1c levels are above 5,6 in the normoglycemic group. This is an important point that should be carefully addressed, given that most comparisons rely on this disbiguation.

Reply: For the diagnosis of prediabetes in terms of fasting glucose and HbA1c, there are some discrepancies: the criteria of the American Diabetes Association are more stringent - the threshold for diagnosing prediabetes is already glycemia 5.6 mmol / l and HbA1c - 5.7%, while WHO criteria are more loyal: value fasting glucose - 6.1 mmol/l, glycated hemoglobin - 6.0% [WHO / IDF, 2021, ADA 2022]. In our study, the WHO criteria were used, which can be considered one of the limitations. At the same time, the criteria for diagnosing diabetes mellitus are the same for the leading medical communities: fasting venous plasma glucose ≥ 7.0 mmol / l (126 mg/dL) on an empty stomach and glucose ≥ 11.1 mmol/l (200 mg/dL) during OGTT or random definition, glycated hemoglobin (HbA1c) 6.5% and more [WHO / IDF, 2021, ADA 2022]. Accordingly, we supplemented the Study Limitations section.

Figure 2. Minor point - Something is wrong with the footnote (part of it is at the end of the page) instead after figure

Reply: We have corrected the footnote text

Figure 2. Measurement of data dispersion should be included in the figure, alongside p values (please apply this to all other figures as well)

Reply: We have not included data dispersion alongside p values in the figure since these values are presented in tables. In our opinion, the presentation of these data in figures will unnecessarily overload them.

I wonder which factors were considered in regression analysis.

Reply: The model included factors such as glucose, type 2 diabetes mellitus, alternative markers of carbohydrate metabolism, overweight or obesity, left atrial size, LV end-diastolic size, total perioperative parameters (duration of surgery, duration of CPB, number of shunts), preoperative heart rate at rest, medical therapy, including hypoglycemic.

As groups differ very much by sex, BMI, smoking status… I wonder what do results are after adjusting for these.

Reply: Regression analysis took into account these factors, they were included in the model, but did not affect the outcomes

Hypoglycemic therapy (including sglt2i) is a conditio sine qua non for this analysis, in my opinion for obvious reasons.

Reply: Medical therapy, including hypoglycemic therapy, was tested in logistic regression, but did not show an association with the studied outcomes

Overall, given that most of the reported correlations concerning primary endpoint, as well as logistic regression imply only weak correlation (barely even that), alongside the fact that not all factors were considered, I would be very careful in concluding based on this data. Hence, discussion section must clearly adresss the weakness of these results, and lack of added clinical benefit. My opinion that the idea for the study is valid, yet it was not well-executed. The observed increase in patients with complications is probably a marker of higher cortisol response in this group.

Reply: Indeed, the logistic regression data reveal a relatively weak correlation, however, despite this, an association of fructosamine with a combined endpoint was revealed, which is quite noteworthy. We have included this fact of weak correlation in the Limitations of the Study section. We believe that the very fact of the association of fructosamine levels and immediate results of CABG may indicate the possible use of this marker in the preoperative evaluation of patients before CABG. In the present study, we did not assess the level of cortisol in patients, so we cannot somehow evaluate the assumption that a high cortisol response is associated with the presence of CABG complications.

Reviewer 4 Report

Thank you for allowing me to review this interesting manuscript. However, I have some minor comments that, if considered, the paper would be improved. 

1. However, this is a secondary analysis paper, and more information about the inclusion and exclusion criteria, sample size calculation, and data collection needs to be mentioned. 

2. You need to state explicitly in the study design that this is a secondary data analysis paper.

3. you need to include clinical implications of your study. 

Author Response

Thank you for allowing me to review this interesting manuscript. However, I have some minor comments that, if considered, the paper would be improved. 

Reply: We would like to thank the reviewer for the work done in reviewing our manuscript and for the favorable feedback. The reviewer's comments helped us improve our manuscript.

  1. However, this is a secondary analysis paper, and more information about the inclusion and exclusion criteria, sample size calculation, and data collection needs to be mentioned. 

Reply: Unfortunately, the manuscript that was provided to you for review lost an important figure with study design, detailed description of inclusion criteria, etc. At present, it has been added (Figure 1), and the text of materials and methods has been supplemented.

  1. You need to state explicitly in the study design that this is a secondary data analysis paper.

Reply: In fact, blood sampling, marker determination and determination of glycemic status and alternative markers of carbohydrate metabolism of patients was carried out during the study in real time, so we cannot claim that this is just a database for secondary data analysis.

  1. you need to include clinical implications of your study. 

Reply: In the text of the manuscript, we discuss the clinical significance of our study, the following paragraph is devoted to this issue:

“What is the possible clinical significance of this study? A recent meta-analysis of 30 studies including 34,650 patients convincingly demonstrated that low glycated hemoglobin (<5.5%) before cardiac surgery is associated with a significant reduction in perioperative complications (such as death, acute kidney injury, neurological and infectious complications) [5]. These data overcome the existing concerns among clinicians about tight perioperative glycemic control [21], and there is a proposal to achieve the maximum possible reduction in the level of glycated hemoglobin before surgery, not only in patients with DM, but in general in all patients before cardiac surgery [22]. However, due to the longer period required to achieve the optimal values of glycated hemoglobin (up to 3 months), such markers of carbohydrate metabolism as fructosamine deserve attention. Until now, their use has been limited due to the lack of convincing data on the possible predictive value of this biomarker. We hope that our study will initiate a more intensive study of this issue”. 

Round 2

Reviewer 2 Report

The authors have addressed most of my concerns. I have no further comments.

Reviewer 3 Report

Dear authors, for start the purpose of figure in scientific paper is not to look "nice", but to reliably represent data. Therefore, by adding the SD, dispersion will be obvious and results more valid, and not "overloaded". Without it, the figure is for magazine rather than scientific paper.

Including so many predictors in regression model is simply not valid from statistical standpoint because of the small sample size.

My general opinion is that conclusions that can be inferred from the present paper are still misleading.

Reviewer 4 Report

Thank you for the effort you made on the revised manuscript.